# Cervicovaginal Microbiota Composition in *Chlamydia trachomatis* Infection: A Systematic Review and Meta-Analysis

**DOI:** 10.3390/ijms23179554

**Published:** 2022-08-23

**Authors:** Marisa Di Pietro, Simone Filardo, Ilaria Simonelli, Patrizio Pasqualetti, Rosa Sessa

**Affiliations:** 1Department of Public Health and Infectious Diseases, University of Rome Sapienza, 00185 Rome, Italy; 2Service of Medical Statistics and Information Technology, Fatebenefratelli Foundation for Health Research and Education, 00100 Rome, Italy; 3Department of Biomedicine and Prevention, University of Rome Tor Vergata, 00133 Rome, Italy

**Keywords:** *Chlamydia trachomatis*, cervicovaginal microbiota, 16s rDNA sequencing, systematic review, meta-analysis

## Abstract

In healthy women, the cervicovaginal microbiota is characterized by the predominance of *Lactobacillus* spp., whereas the overgrowth of anaerobic bacteria leads to dysbiosis, known to increase the risk of acquiring genital infections like *Chlamydia trachomatis*. In the last decade, a growing body of research has investigated the composition of the cervicovaginal microbiota associated with chlamydial infection via 16s rDNA sequencing, with contrasting results. A systematic review and a meta-analysis, performed on the alpha-diversity indices, were conducted to summarize the scientific evidence on the cervicovaginal microbiota composition in *C. trachomatis* infection. Databases PubMed, Scopus and Web of Science were searched with the following strategy: “Chlamydia trachomatis” AND “micro*”. The diversity indices considered for the meta-analysis were Operational Taxonomic Unit (OTU) number, Chao1, phylogenetic diversity whole tree, Shannon’s, Pielou’s and Simpson’s diversity indexes. The search yielded 425 abstracts for initial review, of which 16 met the inclusion criteria. The results suggested that the cervicovaginal microbiota in *C. trachomatis*-positive women was characterized by *Lactobacillus iners* dominance, or by a diverse mix of facultative or strict anaerobes. The meta-analysis, instead, did not show any difference in the microbial biodiversity between *Chlamydia*-positive and healthy women. Additional research is clearly required to deepen our knowledge on the interplay between the resident microflora and *C. trachomatis* in the genital microenvironment.

## 1. Introduction

The cervicovaginal micro-environment harbours a variety of resident microorganisms, named microbiota, whose interplay with the host is involved in either health or disease [1]. The cervicovaginal microbiota is the simplest microbial community in the human organism since, in healthy reproductive-age women, it is characterized by the predominance of one or a few bacterial species, mainly *Lactobacillus* spp., alongside other bacterial species present in much lower amounts, including *Staphylococcus* spp., *Streptococcus* spp., *Bifidobacterium* spp., *Ureaplasma* spp., *Mycoplasma* spp., *Veillonella* spp., etc. [1,2] Indeed, *Lactobacillus* spp. is generally regarded as the main host-defence factor within the cervicovaginal ecosystem against potential pathogens via several mechanisms, including competitive exclusion, anti-microbial compound production (e.g., lactic acid), immune system activation as well as the maintenance of a low vaginal pH [3,4,5]. In certain conditions, the depletion of Lactobacilli and the overgrowth of anaerobic bacteria, like *Gardnerella* spp., *Prevotella* spp., *Atopobium* spp., etc., has often been associated with cervicovaginal dysbiosis, known to increase the risk of acquiring genital infections like *Chlamydia trachomatis* [6,7].

*C. trachomatis* is an obligate intracellular pathogen and leading cause of bacterial sexually transmitted diseases, with more than 128 million new cases per year according to the most recent World Health Organization (WHO) estimates [8,9]. Chlamydial genital infection in women is responsible for urethritis, cervicitis, and salpingitis, although the majority of infections are asymptomatic (>80%) and, thus, untreated, potentially leading to severe reproductive sequelae, including pelvic inflammatory disease, ectopic pregnancy, and obstructive infertility [6,10]. In addition, *C. trachomatis* can be transmitted to infants following the direct contact with infective cervical secretions during delivery, resulting in neonatal conjunctivitis and pneumonia [10]. Lastly, there is evidence that *C. trachomatis* infection increases the risk for Human Immunodeficiency Virus infection and transmission by 3 to 4 times, and, more recently, it has been associated with Human Papillomavirus (HPV)-related cervical cancer [11,12].

In the last decade, high-throughput culture-independent techniques based on the analysis of 16s rRNA gene sequences have significantly contributed to characterizing the composition of the cervicovaginal microbiota in health and disease, leading to the identification of different bacterial profiles [1,13]. In particular, five Community State Types (CSTs), from I to V, have been described, with the CST-I, -II, -III and -V dominated by *Lactobacillus crispatus*, *Lactobacillus gasseri*, *L. iners* and *Lactobacillus jensenii*, respectively, and the CST-IV by a diverse mix of anaerobic bacteria, like *Gardnerella vaginalis*, *Atopobium vaginae* and *Prevotella* spp. CST-IV is often associated with the condition of cervicovaginal dysbiosis, frequently observed in bacterial vaginosis [13].

Since the first attempts, an increasing body of research, investigating the composition of cervicovaginal microbiota associated with *C. trachomatis* genital infection via 16s rRNA gene sequencing, has been published [14,15,16,17,18,19,20,21,22,23,24,25,26,27,28,29]. The main findings evidenced the association of a microbiota dominated by *L. iners* (CST-III) or by different anaerobic bacteria (CST-IV) with *C. trachomatis* infection, although the specific bacterial species identified varied significantly amongst the different studies and, hence, it is very challenging to compare and summarize results, often leading to controversy.

Therefore, the aim of the present systematic review is to provide an update of the scientific evidence on the complex interplay between the cervicovaginal microbiota and *C. trachomatis*, evaluated by epidemiological studies on humans. Moreover, a meta-analysis was performed on the alpha-diversity indices extracted from the included studies.

## 2. Methods

This systematic review and metanalysis has been performed in accordance with the latest version of the Preferred Reporting Items for Systematic Review and Meta-Analysis (PRISMA) guidelines [30]. The Review protocol was registered on the International prospective register of systematic reviews (PROSPERO, reference number CRD42022341268). Zotero citation management software (RRID:SCR_013784) was used to identify any duplicates and to manage and screen the selected literature records.

### 2.1. Literature Research

The Review included articles published up to 30 June 2022, on the databases PubMed, Scopus, and Web of Science, with the following search strategy and Boolean operator: “Chlamydia trachomatis” AND “micro*”. Truncation filters (*) were used to represent any combination of letters. Three independent reviewers (MDP, SF, and IS) performed the search, reading the titles and abstracts of the articles identified by the search strategy. During the multi-step exclusion process, any disagreement on the studies was discussed until reaching a consensus. The process was supervised by other investigators (PP and RS). Figure 1 shows the PRISMA flow chart diagram summarizing the selection steps for the present Systematic Review.

### 2.2. Inclusion and Exclusion Criteria

The review included only studies investigating the composition of the cervicovaginal microbiota via 16s rRNA gene sequencing, comparing reproductive-age women positive to *C. trachomatis* infections to age-matched healthy controls. We did not consider studies analysing the cervicovaginal microbiota in women positive to other genital pathogens, including *Neisseria gonorrhoeae*, *Trichomonas vaginalis*, *Mycoplasma* spp., *Candida* spp., HPV, and herpes simplex virus 2 (HSV-2).

Only articles presenting controlled trials published in peer-reviewed journals were considered eligible. Reviews, meta-analysis, editorials, commentaries, case reports, case series, semi-experimental and experimental studies, proceedings, individual contributions (e.g., conference speeches), and purely descriptive studies published in scientific conferences without any quantitative or qualitative findings, were excluded from the review. Finally, articles published in languages other than English were also excluded. The search was performed until 30 June 2022 on the databases.

### 2.3. Data Extraction Process and Quality Assessment

From each study included in the review, the following data were extracted: bibliographic information, study design, study population size, age, and ethnicity, 16s rRNA gene primers and sequencing platform used, and main conclusions.

Three different reviewers (MDP, SF, and IS) assessed the methodological quality of the selected studies with the Newcastle-Ottawa Scale (NOS) rating tool, adapted for evaluating case-control, cross-sectional and cohort studies [31]. The NOS is divided in eight categories evaluating three different quality aspects: selection, comparability, and outcome; scores range from 0 to 9 and the quality of a study was considered to be high if the NOS score was 7 to 9, intermediate if the NOS score was 4 to 6, and low if it was 0 to 3 [32].

### 2.4. Meta-Analysis

#### 2.4.1. Data Items

The outcomes of interest were the OTU, Chao1, Phylogenetic Diversity whole tree, Shannon’s index, Pielou’s evenness index, and Simpson’s index mean levels in *C. trachomatis* positive (POS) as compared to *C. trachomatis* negative (CTRL) women.

#### 2.4.2. Effect Measure

For each outcome, an effect measure was calculated as the Mean Difference (MD) between the mean values observed in the POS and CTRL groups.

#### 2.4.3. Synthesis Methods

In each synthesis, only the studies that reported the necessary data for MD calculation (mean and SD in each group of interest) were included.

Each parameter of interest showed a high variability; therefore, before the synthesis, difference in means between POS and CTRL groups and the corresponding standard error (SE), calculated on the raw scale, were converted to an approximate difference and standard error on the logarithmic scale following method 3 proposed by Higgins et al. (2008) [33]. To obtain the pooled MD, it was decided to apply the random effects model. MD was reported with the corresponding 95% Confidence Interval (95% CI).

The results were represented by the forest plot that is a graphical representation of the estimated results from each study included in the analysis along with the pooled result. The heterogeneity between the studies was assessed through the visual inspection of the forest plot, by the overlap of the CIs and possible outliers, and quantified through the I^2^ index. The I^2^ statistic describes the percentage of variation across studies that is due to heterogeneity rather than chance. We considered that an I^2^ of 50% indicates moderate heterogeneity and 75% or greater indicates substantial heterogeneity. The *p* value of the chi-squared (χ^2^) test, a statistical test for heterogeneity, was included in the forest plots.

Publication bias was assessed with visual inspection of the funnel plot, that is a scatter plot of the effect estimates from individual studies against standard error (SE). Egger’s test was not performed because in each analysis the number of studies was less than 10. We considered a *p* value < 0.05 to be statistically significant. The data was entered and analysed with the statistical software STATA v16.

## 3. Results and Discussion

### 3.1. Study Selection Process

In total, we recovered 425 studies from all searched databases (*n* = 162 from Scopus, *n* = 165 from Web of Science, and *n* = 98 from PubMed), and after applying filters by automation tools, 317 articles remained. Out of the remaining 317 papers, 183 were excluded after removing duplicates, and 134 were subjected to further screening and evaluated for inclusion in the systematic review after considering inclusion and exclusion criteria. A total of 117 papers was then excluded because they did not fit inclusion criteria. One more article was not considered since it did not included controls. At the end of the process, 16 articles were included in the systematic review. Due to difficulties in extrapolating alpha-diversity data, only 7 of them were considered for the meta-analysis.

### 3.2. Characteristics of the Included Studies

The characteristics of the studies included in the Systematic Review are summarized in Table 1. The papers were published between 2017 and 2021 and were conducted on almost all continents, with 8 of them performed in Europe [14,19,21,23,24,25,26,29], 2 in North America [17,20], 3 in Asia [16,22,27], 2 in Africa [18,28] and 1 in Australia [15]. 13 case-control [14,16,17,18,19,22,23,24,25,26,27,28,29], 2 cohort [15,20] and 1 cross-sectional [21] studies were included, and all of them involved reproductive-age women, who tested positive for *C. trachomatis* infection via nucleic acid amplification tests (NAATs), alongside matched uninfected women as control group. In addition, 3 studies followed *C. trachomatis*-positive women after standard azithromycin treatment [15,20,22], 3 studies investigated women with a *C. trachomatis* co-infection with HPV [29], *Mycoplasma genitalium* [17], or *T. vaginalis* and/or *N. gonorrhoeae* [16], and 2 studies compared the vaginal microbiota in *C. trachomatis*-positive women to that in women with *T. vaginalis* or *C. albicans* genital infections [26,28]. 15 studies performed the 16s rDNA sequencing via Illumina platform, whereas one study employed the Ion torrent PGM platform [28]. Amongst them, 13 studies sequenced the hypervariable region V3-4 [14,15,16,17,19,20,21,22,23,24,26,27,29], 2 studies the V4 [18,25] and 1 study the V2-4-8 region [28]. The majority of the included studies (*n* = 9) collected vaginal swabs [15,16,17,19,20,21,22,26,28], whereas 4 analysed endocervical swabs [24,25,27,29], 2 endocervical and/or vaginal swabs [14,18], and one study compared vaginal swabs to anal swabs from women with a concomitant *C. trachomatis* anorectal infection [23]. The sample size was very variable, ranging from 7 cases and 7 controls in the smallest case-control study [24], to a cohort of 248 women [20].

### 3.3. Cervicovaginal Composition in C. trachomatis-Positive Women

The majority of the studies included in our review evidenced an overall decrease in the relative abundance of *Lactobacillus* spp., specifically *L. crispatus*, in *C. trachomatis*-positive women as compared to healthy controls [14,16,17,18,19,20,21,22,23,24,25,26,27,28,29]. In particular, a chlamydial genital infection was mostly associated with a cervicovaginal microbiota characterized by either the dominance of *L. iners*, or a diverse mix of facultative or strict anaerobes, with the most frequently identified being *G. vaginalis, A. vaginae*, *Megasphaera* spp., *Prevotella* spp., *Parvimonas* spp. [14,16,17,18,19,20,21,22,23,24,25,26,27,28,29]. These bacterial species are usually the hallmark of a CST-IV microbiota, a condition often associated with bacterial vaginosis and believed to increase the risk of acquiring genital pathogens [13,34,35].

From the included studies has also emerged the need to restore a protective vaginal microbiota after antibiotic treatment for *C. trachomatis*, via, for example, the use of probiotics [15,20,22]. In this regard, besides comparing the resident microflora of the genital ecosystem between *C. trachomatis*-positive and healthy women at baseline, some authors have also investigated the effects of the standard treatment with azithromycin. In particular, Ziklo et al. (2018) have observed an increased prevalence of indole-producing bacteria in the vaginal microbiota post-treatment, suggesting a destructive effect of azithromycin on vaginal health [15]. On a similar note, the study from Tamarelle et al., 2020, has described a robust increase in *L. iners*’ abundance, suggesting that after antibiotic treatment, the individual risk for *C. trachomatis* remains high, as evidenced by the increased rate of reinfections observed in their study [22].

Another interesting aspect was the evidence that unique microbial profiles might characterize genital infection with other viral, protozoan, or fungal pathogens, potentially representing distinct markers of infection [16,17,26,28,29]. Indeed, the compositional structure and biodiversity of the cervicovaginal microbiota in *C. trachomatis*-positive women was also compared to women with different co-infections, or to women with other genital pathogens alone. The main findings showed that mixed infections, with *C. trachomatis* and HPV, *M. genitalium*, and *N. gonorrhoeae*/*T. vaginalis*, are associated with a CST-IV microbiota, with increased anaerobic bacterial species, or to *L. iners*-dominated microbiota, and did not present significant differences as compared to women positive to *C. trachomatis* alone [16,17,29]. By contrast, women with HPV alone showed a cervicovaginal microbiota similar to that observed in healthy controls, whereas women with either *T. vaginalis* or *C. albicans* infections presented a microbiota characterized by a lower abundance of *Lactobacillus* spp. and much higher abundance of anaerobes, such as *Prevotella* spp., *Gardnerella* spp., *Atopobium* spp. and *Megasphaera* spp., as compared to *C. trachomatis*-positive women [26,28].

### 3.4. Meta-Analysis

Meta-analysis was performed using the following indices: Chao1’s diversity index, OTU number, phylogenetic diversity whole tree, Shannon’s diversity index, Pielou’s evenness diversity index, and Simpson’s diversity index.

#### 3.4.1. Chao1’s Diversity Index

Four studies were eligible for the synthesis of Chao1 mean levels for a total of 95 subjects, 43 positive and 52 controls. The results of the meta-analysis indicate a non-significant difference between positives and controls in Chao1 levels; the pooled MD on a logarithmic scale was equal to −0.06 (95% CI −0.33 to 0.20; *p* = 0.638) corresponding to an approximate relative reduction in the positives group of about −6%. The heterogeneity between the studies was not significant (I^2^ = 51.1%; *p* = 0.105) (Figure 2A). All the studies presented a NOS score of 8 and only one of 7 [22] (Table 2). The funnel plot seemed to not show an evidence of publication bias even if there were only 4 studies included in the analysis (Appendix A).

#### 3.4.2. OTU Number

Four studies were included in the meta-analysis of OTU levels. Pooling the studies there was a total of 207 subjects, 72 positive and 135 controls. The results of the meta-analysis showed a non-significant difference between positives and controls in the OTU; the pooled MD on a logarithmic scale was equal to 0.11 (95% CI −0.28 to 0.50; *p* = 0.582) corresponding to a relative increase in positives of approximately 11% (Figure 2B). The heterogeneity between the studies was high and significant (I^2^ = 86.4%; *p* < 0.001). In fact, two studies [22,26] reported a negative difference indicating lower values in positive group, while the other two studies reported a positive difference indicating higher values in the positive group [24,25]. All the studies presented a NOS score of 8 and only one of 7 [22] (Table 2). The funnel plot does not seem to show the presence of publication bias but there were only 4 studies (Appendix A).

#### 3.4.3. Phylogenetic Diversity Whole Tree

Only two studies reported data about this parameter [24,26] so the analysis was performed considering these two studies. The pooled total number of subjects was of 55, 27 positives and 28 controls. The results of the meta-analysis indicate a non-significant difference between positives and controls; the pooled MD on a logarithmic scale was equal to 0.08 (95% CI −0.12 to 0.28; *p* = 0.412) corresponding to a relative increase in the positive group of about 8% (Figure 2C). The heterogeneity between the studies was not significant (I^2^ = 26.8%; *p* = 0.242) (Appendix A).

#### 3.4.4. Shannon’s Diversity Index

Seven studies report data for Shannon’s parameter for a total of 345 subjects, comprising 138 positives and 207 controls. The results of the meta-analysis indicated a non-significant difference between positives and controls; the pooled MD on a logarithmic scale was equal to 0.21 (95% CI −0.08 to 0.51; *p* = 0.158) corresponding to a relative increase in the positive group of about 21%. The heterogeneity between the studies was significant (I^2^ = 77.1%; *p* < 0.001) (Figure 2D). Two studies [24,25] reported a positive and significant difference, while the remaining 5 studies reported non-significant results. The studies included in this meta-analysis were of high quality except for one of intermediate quality [27] (Table 2). The funnel plot did not seem to show the presence of publication bias (Appendix A).

#### 3.4.5. Simpson’s Diversity Index

Four studies were eligible for the meta-analysis of the Simpson parameter [22,24,27,28]; a total number of 140 subjects were considered, comprising 69 positives and 71 controls. The results of the meta-analysis indicate a non-significant difference between positives and controls. The pooled MD on a logarithmic scale was equal to 0.09 (95% CI −0.40 to 0.58; *p* = 0.719) corresponding to a relative increase in the positives of about 9% (Figure 2E). The heterogeneity between the studies was significant (I^2^ = 70.4%; *p* = 0.017). In this meta-analysis, the MD observed in Filardo et al., 2017 [24] study was positive and significant, while in the other three studies the results were not significant. Based on NOS score, all the studies were of high quality with the exception of Cheong et al. (2019) [27] of intermediate quality (Table 2). The funnel plot did not seem to show evidence of publication bias (Appendix A).

#### 3.4.6. Pielou’s Evenness Diversity Index

Three studies report data about the parameter Pielou [23,25,27] for a total of 241 subjects, comprising 91 positives and 150 controls. The results of the meta-analysis showed a non-significant difference between positives and controls. the pooled MD on a logarithmic scale was equal to 0.21 (95% CI −0.15 to 0.58; *p* = 0.250), corresponding to a relative increase in positives of about 21% (Figure 2F). The heterogeneity between the studies was significantly high (I^2^ = 80.9%; *p* = 0.005). All the studies included in the meta-analysis were of high to very high quality, with the exception of one study [27] of intermediate quality (Table 2). It was difficult to exclude the presence of publication bias by the funnel plot since there were only 3 studies included in the analysis (Appendix A).

### 3.5. Scoring Results

The median NOS score of the included studies was 8 (interquartile range, IQR 1.25), indicating a high average quality level. Specifically, most studies (*n* = 12) were of high to very high quality (score of 7 to 9), 3 studies were of intermediate quality (score of 4 to 6), and 1 study was of low quality (score of 0 to 3). Table 2 shows the results of the scoring method applied to each study included in the review, with reference to publication year and study design.

## 4. Conclusions

Overall, based on the findings presented in our review, it emerged that women affected by a *C. trachomatis* infection possessed an altered cervicovaginal microbiota, with increased abundances of a varied population of strict or facultative anaerobic bacterial species as well as of *L. iners*.

Concerning the quantitative measures of microbial biodiversity within a bacterial community, namely alpha- diversity indices, a non-significant difference was demonstrated, via a meta-analysis, between *C. trachomatis* positive and healthy women. Indeed, alpha-diversity indices were scarcely used in the reviewed studies; only about half of them reported alpha-diversity data via a plethora of different indices, including Shannon, Shannon–Weaver’s or Pielou’s evenness, Chao1, Simpson’s, etc., leading to inconsistent results [23,24,25,26,27,28,29]. Specifically, some papers did not find any statistically significant difference in the diversity indices between *C. trachomatis*-positive women and healthy controls [26,27,28], while others did find increased diversity in the cervicovaginal microbiota of *Chlamydia*-infected women [23,24,25,29].

There are different limitations that undermine the comparability of results in the studies included in our systematic review and meta-analysis. In particular, there are differences in specimen collection, since either vaginal and/or endocervical swabs have been used; indeed, it is known that significant differences have been demonstrated in the microbiota characterizing the two sites. In addition, different primers and hypervariable regions of 16s rDNA were chosen for sequencing, such as the V3-4, the V4 or the V2-4-8, leading to potentially different results for microbiota composition. In fact, it has been already evidenced that different primers may influence the results of the bioinformatic analysis. At the same time, another relevant issue lies in the statistical analysis since a varied mix of statistical algorithms have been used. In this regard, it is very challenging to compare the biodiversity of the genital microbiota between different studies and patient groups, due to the numerous and diverse statistical measures adopted for the calculation of alpha-diversity indices, as well as to missing or incomplete reporting of data in the papers.

In the future, it will be of great interest to reach a consensus on the several parameters involved in designing metagenomic studies, so that all data are reported and rendered available to the scientific community. Indeed, given the importance of the microbiota not only from a pathophysiological but also clinical point of view, streamlining the presentation of metagenomic data will render their interpretation more accessible to a wider audience. Surely, a standardized procedure will help to significantly push the field forward by providing more complete data for a better understanding of the pathophysiological role of the cervicovaginal microbiota in *C. trachomatis* infection.

## Figures and Tables

**Figure 1 ijms-23-09554-f001:**
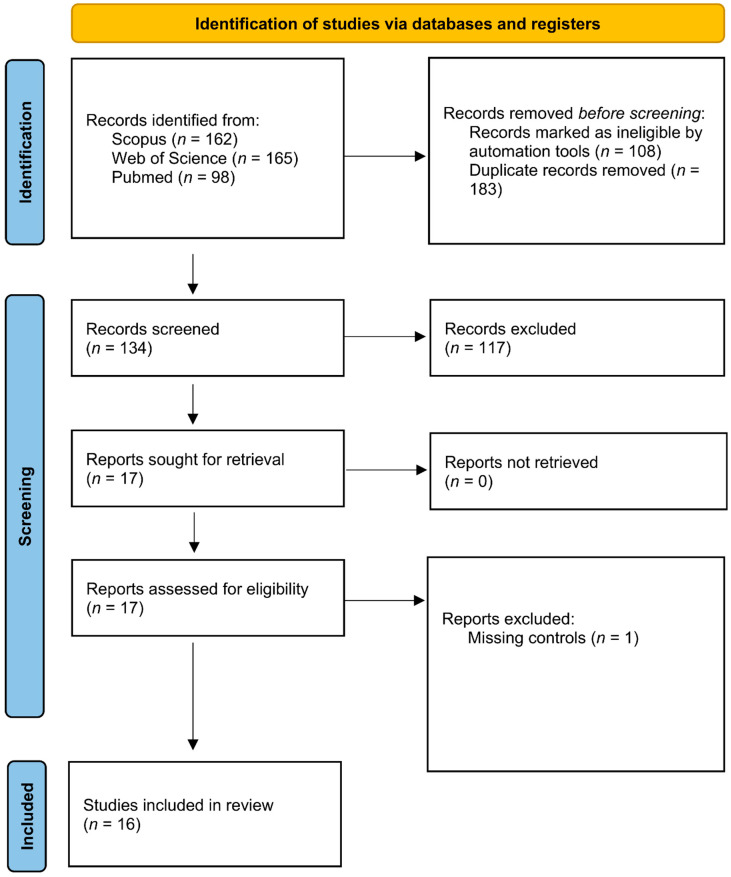
PRISMA flow diagram.

**Figure 2 ijms-23-09554-f002:**
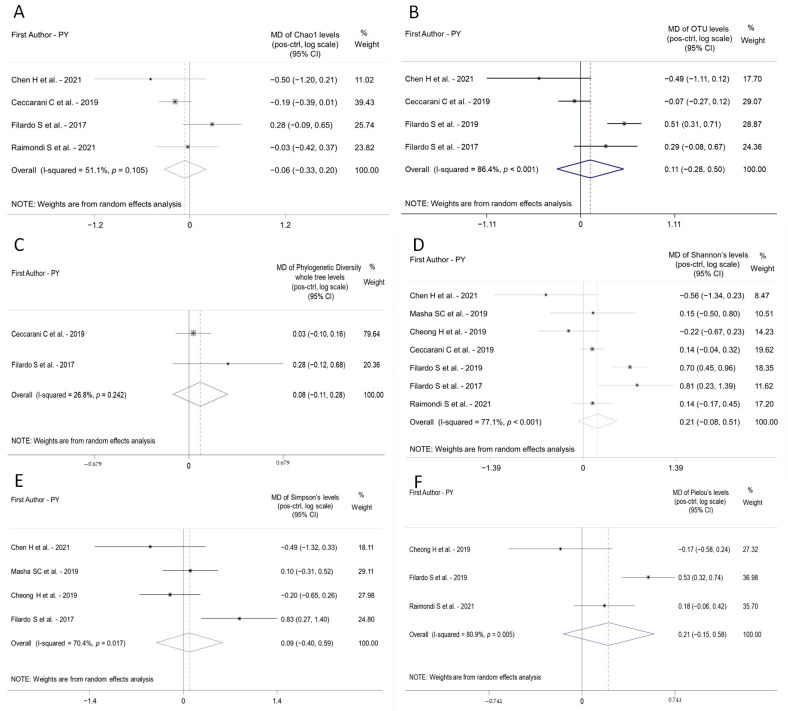
Forest plot displaying the results of the meta-analyses of studies reporting data about each Alpha-Diversity parameter in the positive group (pos) and healthy controls (ctrl). (**A**), Chao1; (**B**), OTU number; (**C**), Phylogenetic diversity whole tree; (**D**), Shannon; (**E**), Simpson; (**F**), Pielou’s evenness). The point estimated by each study (Mean Difference on logarithmic scale between pos and ctrl, MD) is represented by a square and the horizontal line is the 95% Confidence Interval (95% CI). The area of the square reflects the weight that the study contributes to the meta-analysis. The combined-effect MD and its 95% CI are represented by the diamond. The vertical line represents the line of no difference. CI, confidence interval; MD, Mean Difference; PY, Publication Year. See Refs. [22,23,24,25,26,27,28].

**Table 1 ijms-23-09554-t001:** Characteristics of the studies (*n* = 16) included in the systematic review.

Authors	Study-Period	Study Population	Ethnicity	Sample Type	Sequencing Platform	Main Results
Filardo et al. [24]	2016	Women with *C. trachomatis* (CT, *n* = 7), and healthy controls (HC, *n* = 7)	European	Endo-cervical swabs	V3-4 Illumina	CT women showed a marked increase in alpha-diversity indices (Shannon’s and Shannon-weaver’s) and an overall decrease in *Lactobacillus* spp., alongside an increase in anaerobic bacterial species, including *G. vaginalis*, *A. vaginae*, *P. amnii*, *P. timonensis*, and *L. amnionii*.
van der Veer et al. [14]	2013–2014	Women notified for *C. trachomatis* (CT) infection of sex partner (*n* = 93), of which *n* = 52 tested CT positive and *n* = 41 tested CT negative (controls)	European	Endocervical and/or Vaginal swabs	V3-4 Illumina	CT women were significantly associated with a cervico-vaginal microbiota characterized by diverse anaerobic bacteria or with a microbiota dominated by *L. iners*, as compared to a microbiota dominated by *L. crispatus*.
Balle et al. [18]	2013–2014	Women with *C. trachomatis* (CT+, *n* = 30), and uninfected controls (CT-, *n* = 42)	African	Endocervical and vaginal swabs	V4 Illumina	The endocervical microbiota diversity is not grossly altered in CT+ women, although CT+ women had higher relative abundance of *G. vaginalis* and other anaerobes, like *Megasphaera* spp., *A. vaginae*, *Dialister* spp., and *Prevotella* spp., all BV-associated bacteria, as compared to CT- women.
Ziklo et al. [15]	Not reported	Women with diagnosed *C. trachomatis* infection at baseline (CT-P, *n* = 11), with repeated CT infection in the last year (CT-RP, *n* = 3) and post antibiotic treatment (PAT, *n* = 13), as well as CT-negative controls (CT-N, *n* = 10)	Australian	Vaginal swabs	V3-4 Illumina	CT-P and CT-RP women were associated with elevated vaginal kynurenine/tryptophan ratios. CST-IV (anaerobic bacteria) showed significantly lower vaginal tryptophan levels as compared to CST-I (*L. crispatus*) and III (*L. iners*). In PAT women, a higher abundance of indole producing bacterial species were observed, including *Porphyromonas asaccharolytica*, *Propionibacterium acnes*, *Fusobacterium nucleatum*, *Faecalibacterium prausnitzii*, *Enterococcus faecalis*, *Peptoniphilus harei*, and *Escherichia coli*.
Tamarelle et al. [21]	2015	Women with *C. trachomatis* (CT+, *n* = 21), and uninfected controls (CT-, *n* = 111)	European	Vaginal swabs	V3-4 Illumina	CSTs were not significantly associated with *C. trachomatis* status but higher proportions of CT+ women were found in CST-III (*L. iners*) and CST-IV (anaerobic bacteria), rather than in CST-I (*L. crispatus*).
van Houdt et al. [19]	2008–2012	Women screened for *C. trachomatis* who tested negative at the first visit (*n* = 115). At subsequent yearly screening, *n* = 60 women tested CT-positive, and *n* = 55 women tested CT-negative (controls).	European	Vaginal swabs	V3-4 Illumina	Five CSTs were identified, four CSTs were dominated by *Lactobacillus* spp., of which *L. crispatus (CST-I)* and *L. iners (CST-III)* were the most common, and one CST was characterized by an array of strict and facultative anaerobes (CST-IV). Women with *L. iners* dominated CST-III had increased risk of CT infection.
Di Pietro et al. [29]	2016	Women with *C. trachomatis* infection (CT, *n* = 10), papilloma virus infection (HPV, *n* = 10), HPV/CT co-infection (*n* = 5), and healthy controls (HC, *n* = 10).	European	Endocervical swabs	V3-4 Illumina	Alpha diversity indices (Shannon’s and Shannon-weaver’s) were higher in either CT or HPV/CT co-infected women as compared to healthy controls. The cervical microbiota of CT positive and HPV/CT co-infected women was characterized by decreased *Lactobacillus* spp. and increased anaerobic bacterial species, like *G. vaginalis*, *A. vaginae* and *A. christensenii*. *L. iners* were also more frequently found in CT positive and HPV/CT co-infected women. By contrast, HPV positive women showed a similar microbiota to those in healthy controls.
Masha et al. [28]	2015	Pregnant women with *Trichomonas Vaginalis* (TV, *n* = 18), compared to pregnant women with *C. trachomatis* (CT, *n* = 14), and healthy controls (HC, *n* = 21)	African	Vaginal swabs	V2-4-8 Ion torrent PGM	Bacterial alpha-diversity indices (Simpson’s and Shannon’s) were significantly higher in women with either TV or CT as compared to healthy controls. Women with TV had increased abundance of *Parvimonas* and *Prevotella* spp. as compared to both CT+ women and healthy controls, whereas CT+ women had increased abundance of *Anaerococcus*, *Collinsella*, *Corynebacterium*, and *Dialister* spp.
Cheong et al. [27]	2010–2014	Women with *C. trachomatis* infection (CT, *n* = 42), and healthy controls (HC, *n* = 35)	Asian	Endocervical swabs	V3-4 Illumina	Women with CT infection showed no increased cervical bacterial alpha-diversity indices (Simpson’s, Shannon’s and Pielou’s). CT infection was associated to increased abundances of strict and facultative anaerobes, like *Streptococcus, Megasphaera, Prevotella* and *Veillonella* spp.
Ceccarani et al. [26]	2016	Women with bacterial vaginosis (BV, *n* = 20), vulvovaginal candidiasis (VVC, *n* = 18), C. trachomatis (CT, *n* = 20), and healthy controls (HC, *n* = 21)	European	Vaginal swabs	V3-4 Illumina	Alpha-diversity indices (Shannon’s and Chao1) were higher in BV women as compared to CT and HC women. *Lactobacillus* spp. were decreaed In BV, VVC and CT groups, while HC group microbiota was dominated by *L. crispatus. In* BV, VVC and CT, *L. crispatus* was replaced by *L. iners.* CT, BV and VVC, were characterized by anaerobes, such as *Gardnerella*, *Prevotella*, *Megasphaera*, *Roseburia* and *Atopobium* spp. The decrease of lactate was considered as a common marker of all the pathological conditions.
Filardo et al. [25]	2017	Women with *C. trachomatis* infection (CT, *n* = 42), and healthy controls (HC, *n* = 103).	European	Endocervical swabs	V4 Illumina	Alpha-diversity indices (Shannon’s and Shannon–Weaver’s) were significantly higher in CT women as compared to HC. CT microbiota was dominated by anaerobes (CTS-IV), and a specific network of *G. vaginalis*, *P. amnii*, *P. buccalis*, *P. timonensis*, *A. christensenii* and *V. guangxiensis* was identified as potential biomarker of CT infection. CT was also significantly correlated with increased levels of lactoferrin, IL-6, IL-1α, IFN-α, and IFN-β, whereas very low levels of IFN-γ were observed.
Borgogna et al. [17]	Not reported	Women with *C. trachomatis* infection (CT+, *n* = 54), CT/*Micoplasma genitalium* co-infection (CT+/MG+, *n* = 14), and healthy controls (HC, *n* = 77)	African-American	Vaginal swabs	V3-4 Illumina	Women with CT infection or coinfection CT/MG were associated with a CST-IV microbiota, characterized by decreased *Lactobacillus* spp. Significant differences in vaginal metabolites were identified in CT+ or CT+/MG+ women as compared to uninfected women, before and after adjustment for CSTs, with significant overlap between CT+ and CT+/MG+ women.
Tamarelle et al. [20]	Not reported	Women with confirmed *C. trachomatis* infection at baseline and after azithromycin treatment at 3, 6 and 9 months (*n* = 149), and CT negative controls (*n* = 99).	African-American	Vaginal swabs	V3-4 Illumina	CT women microbiota was dominated, at the time of diagnosis, by *L. iners* or a diverse array of BV or CST-IV associated bacteria, such as *G. vaginalis*, *A. vaginae* and *M. curtisii*. *L. iners*-dominated communities were most common after azithromycin treatment (1 g monodose), consistent to the observed relative resistance of this bacterium to azithromycin.
Chen et al. [22]	2019–2020	Women with tubal infertility and *C. trachomatis* infection before (CT-P, *n* = 6) and after treatment (CT-PT, *n* = 4), as compared to infertile (CT-N, *n* = 8) or healthy women (CT-C, *n* = 7) without chlamydial infection (controls)	Asian	Vaginal swabs	V3-4 Illumina	Women with tubal infertility and CT presented a *L. iners* dominated microbiota with a decrease in *Lactobacillus, Bifidobacterium*, *Enterobacter*, *Atopobium,* and *Streptococcus* spp., which could be restored with varying degrees by azythromycin treatment. *C. trachomatis*-positive women also had increased levels of IFNγ and IL-10.
Raimondi et al. [23]	2019	Women with contemporary vaginal and ano-rectal *C. trachomatis* infection (CT-positive, *n* = 10), and uninfected controls (CT-negative, *n* = 16).	European	Vaginal and anal swabs	V3-4 Illumina	Alpha-diversity via Pielou’s index was higher in the vaginal microbiota of CT-positive women than CT-negative women. In CT-positive women, the vaginal microbiota was depleted of *Lactobacillus* spp., with a significant increase in anaerobes, like *Sneathia* spp., *Parvimonas* spp., and *Megasphaera* spp. CT positively correlated with *Ezakiella* spp. The predicted metabolic functions showed increased chorismate and aromatic amino-acid biosynthesis, as well as mixed acid fermentation, in the vaginal microbiota of CT-positive women.
Chiu et al. [16]	2018–2019	Women with vaginitis and *C. trachomatis* (CT, *n* = 22), *Trichomonas vaginalis* (TV, *n* = 7), *Neisseria gonorrhoeae* (GC, *n* = 2)*,* mixed infections (TV/CT, *n* = 2; TV/CT/GC, *n* = 1), as well as uninfected controls (non-STI, *n* = 36).	Asian	Vaginal swabs	V3-4 Illumina	In CT women, the vaginal microbiota was dominated by *L. iners*, with increased relative abundance of *G. vaginalis* as compared to TV and non-STI women. In TV women, *Lactobacillus* spp. was significantly lower, and *S. agalactiae, P. bivia*, *S. sanguinegens* and *G. asaccharolytica* were significantly enriched, as compared to the other patient groups.

**Table 2 ijms-23-09554-t002:** Newcastle-Ottawa scoring (NOS) results of the included studies in relation to the study design.

Authors	Year	Journal	Country	Study Design	NOS Score
Filardo et al. [24]	2017	Front. Cell. Infect. Microbiol.	Italy	Case-control study	8
van der Veer et al. [14]	2017	Clin. Infect. Dis.	Netherlands	Case-control study	8
Balle et al. [18]	2018	Sci. Rep.	South Africa	Case-control study	8
Ziklo et al. [15]	2018	Front. Cell. Infect. Microbiol.	Australia	Cohort study	5
Tamarelle et al. [21]	2018	Sex. Transm. Infect.	France	Cross-sectional study	4
van Houdt et al. [19]	2018	Sex. Transm. Infect.	Netherlands	Case-control study	8
Di Pietro et al. [29]	2018	New Microbiologica	Italy	Case-control study	8
Masha et al. [28]	2019	PLoS One	Kenya	Case-control study	7
Cheong et al. [27]	2019	PLoS One	China	Case-control study	6
Ceccarani et al. [26]	2019	Sci. Rep.	Italy	Case-control study	8
Filardo et al. [25]	2019	mSystems	Italy	Case-control study	8
Borgogna et al. [17]	2020	Sci. Rep.	USA	Case-control study	6
Tamarelle et al. [20]	2020	J. Infect. Dis.	USA	Cohort study	7
Chen et al. [22]	2021	Front. Cell. Infect. Microbiol.	China	Case-control study	7
Raimondi et al. [23]	2021	Pathogens	Italy	Case-control study	8
Chiu et al. [16]	2021	Microorganisms	China	Case-control study	3

## Data Availability

Not applicable.

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
