# Peer review of "Cervicovaginal Microbiota Composition in Chlamydia trachomatis Infection: A Systematic Review and Meta-Analysis"

_ijms, 2022, doi:10.3390/ijms23179554_

Round 1
Reviewer 1 Report
This is a carefully done systematic review of articles comparing the cervicovaginal microbiota in patients with Chlamydia trachomatis to that of healthy controls. The authors identify 16 articles meeting the inclusion and exclusion criteria, collect the relevant information prseneted in them, score the methodology used according to NOS criteria and perform a meta-analysis on various alfa-diversity indices. While most studies point towards women with C. trachomatis infection having an altered cervical microbiota, either with Lactobacillus iners or increased abundance of various strict or facultative anaerobes, the meta-analysis using various quantitative indices showed nonsignificant differences. Comparison is hampered by variabilities in the methodology, including specimen site, 16S primers and statistical analysis of data (an maybe also handling of specimen including collection tube and nucleic acid extraction method).
This reviewer has no major criticism to the methods, results or conclusions of the review, nor references included.
There are some minor inconsistencies in the manuscript: 16 articles are presented in the text as well as in Table 2 while Figure 1 ends up with 15 articles tin the systematic review. Also, the last timepoint covered in the literature search is said to be 31st June 2021 (line 87) or 31st June 2022 (line 109). In both cases the date should probably be 30th June.
Author Response
|
Reviewer 1’s comments
|
Authors’ answers
|
|
There are some minor inconsistencies in the manuscript: 16 articles are presented in the text as well as in Table 2 while Figure 1 ends up with 15 articles tin the systematic review. Also, the last timepoint covered in the literature search is said to be 31st June 2021 (line 87) or 31st June 2022 (line 109). In both cases the date should probably be 30th June.
|
We thank the Reviewer for the positive comments, and we are sorry for the highlighted oversights. We fixed the number of papers included in the systematic review, that we misreported in Table 2, and we corrected the dates, being the right one 30th June 2022.
|
Reviewer 2 Report
Di Pietro et al.
for Intl J Mol Sci
Cervicovaginal Microbiota Composition in Chlamydia trachomatis Infection: a Systematic Review and Meta-Analysis
Summary:
The authors present an overview of the available comparable microbiome studies of child-bearing aged women with and without infection by C. trachomatis. They find that infected women had an altered miocrobiome relative to their uninfected peers.
Comments:
This work carefully pulls together various studies to produce and evaluate the quality of consensus observations. It is well written and appears to be a good fit for IJMS. There are a few grammar or word choice issues, which should be simple for a copy editor to catch and correct. In this vein, do the authors mean Human Immunodeficiency Virus on line 56?
Author Response
|
Reviewer 2’s comments
|
Authors’ answers
|
|
This work carefully pulls together various studies to produce and evaluate the quality of consensus observations. It is well written and appears to be a good fit for IJMS. There are a few grammar or word choice issues, which should be simple for a copy editor to catch and correct. In this vein, do the authors mean Human Immunodeficiency Virus on line 56?
|
Thank you for the positive comments. We did indeed mean Human Immunodeficiency Virus, on line 56; we corrected the oversight. We revised the entire manuscript for grammar or word choice issues.
|